# Metabolism and Biological Activities of 4-Methyl-Sterols

**DOI:** 10.3390/molecules24030451

**Published:** 2019-01-27

**Authors:** Sylvain Darnet, Hubert Schaller

**Affiliations:** 1CVACBA, Instituto de Ciências Biológicas, Universidade Federal do Pará, Belém, PA 66075-750, Brazil; 2Plant Isoprenoid Biology (PIB) team, Institut de Biologie Moléculaire des Plantes du CNRS, Université de Strasbourg, Strasbourg 67084, France

**Keywords:** sterol, C4-demethylation complex (C4DMC), 4-methylsterol, hormone, steroid, development, genetic disease

## Abstract

4,4-Dimethylsterols and 4-methylsterols are sterol biosynthetic intermediates (C4-SBIs) acting as precursors of cholesterol, ergosterol, and phytosterols. Their accumulation caused by genetic lesions or biochemical inhibition causes severe cellular and developmental phenotypes in all organisms. Functional evidence supports their role as meiosis activators or as signaling molecules in mammals or plants. Oxygenated C4-SBIs like 4-carboxysterols act in major biological processes like auxin signaling in plants and immune system development in mammals. It is the purpose of this article to point out important milestones and significant advances in the understanding of the biogenesis and biological activities of C4-SBIs.

## 1. An Introduction to 4-Methylsterols

Post-squalene sterol biosynthesis consists in the enzymatic conversion of C_30_H_50_O steroidal triterpene precursors such as lanosterol or cycloartenol into pathway end-products among which the most popular are cholesterol, ergosterol, poriferasterol, sitosterol, and many others distributed among eukaryotes. Several dozens of sterol structures may be detected and identified in given organisms or tissues [1,2,3,4,5]. Biosynthetic relationships between all these sterol structures have been extensively documented [6,7,8]. Sterol structural differences between eukaryotic kingdoms involve the number of exocyclic carbon atoms at position C24 and unsaturations in the B cycle of the cholestane backbone (Figure 1A,B). Cholesterol is a Δ^5^-sterol bearing the eight carbon side chain at position C17, which is a structural feature resulting from the cyclization of 2,3-oxidosqualene (C_30_H_50_O) into a protosteryl cationic reaction intermediate and then into lanosterol or cycloartenol [9]. In plants, campesterol and sitosterol are Δ^5^-sterols with one and two methyl groups at position C24, respectively. In yeast, ergosterol is a Δ^5,7^-sterol with one extra methyl group at C24. Sterol pathways are markedly different between eukaryotes depending on the cyclization of 2,3-oxidosqualene into lanosterol in fungi and mammals or cycloartenol in some protists and plants (Figure 2). In fact, this dichotomy generates the particular series of 9β,19-cyclopropylsterols derived from cycloartenol, the biosynthetic and functional features of which have been discussed [6,7,8].

The enzymatic conversion of lanosterol or cycloartenol into pathway end-products (cholesterol, ergosterol, and phytosterols) implies crucial demethylation steps at C14 and C4 positions. Here again, substrates of these reactions in the eukaryotic kingdom differ. Mammals and fungi perform two consecutive C4-demethylations of 30-nor-lanosterol occurring right after the mandatory C14-demethylation of lanosterol, whereas plants carry out two distinct and nonconsecutive C4-demethylations, the first one applying to a 4,4,14-trimethylsterol and the second one to a 4,14-dimethylsterol or a 4-methylsterol (Figure 2).

Both sterol demethylations at C4 and C14 require molecular oxygen for the oxidative cleavage of carbon-carbon bonds, but enzymes at play are different. Demethylation at C14 is catalyzed by a 14α-methylsterol-14α-methyl-demethylase, which is a cytochrome P450—dependent mono-oxygenase also known as CYP51 in mammals [10,11,12], in yeast [13] and in plants [14,15,16]. A Δ^14^-sterol-14-reductase catalyzes the reduction of the resulting Δ^8,14^-diene (Figure 2 and Appendix A). This enzyme is encoded by a single gene in plants and yeast [17,18], while in human two distinct genes were characterized [19,20]. In the same organisms, the demethylation at C4 leads to the production of 4α-carboxysterols by an oxygen-dependent process followed by an oxygen-independent C-C cleavage that generates 3-ketosterols (Appendix A) [21]. It is now established that sterol-C4-demethylation implies the consecutive action of three enzymes: a sterol-4α-methyl oxidase (SMO), a 3β-hydroxysteroid dehydrogenase/C4-decarboxylase (C4D), and a sterone ketoreductase (SKR) [22] (Figure 2 and Appendix A). A protein called ERG28 was shown to tether all three enzymes as a complex in the endoplasmic reticulum [23].

A prominent category of sterol biosynthetic intermediates is 4-methylsterols (including 4,4-dimethylsterols) hereafter collectively named C4-Sterol Biosynthetic Intermediates (C4-SBIs). These molecules with one or two methyl groups at position C4 are precursors of 4-desmethylsterols, like cholesterol in animals, ergosterol in yeast, poriferasterol in some algae, and phytosterols in plants, as stated above. 

The C4-SBIs in mammals and fungi are compounds which follow each other in a biosynthetic segment joining lanosterol to zymosterol. In plants, the pathway is different. The C4-SBIs include compounds which succeed each other in biosynthetic segments joining cycloartenol to episterol or Δ^7^-avenasterol, two 24-alkyl-4-desmethylsterols (Figure 2). C4-SBIs are amphiphilic molecules with a rigid structure just like 4-desmethylsterols. Four rings (A, B, C, D) form a quasi-planar tetracyclic nucleus, with a hydroxy or keto group at the C3 position, one or two methyl groups at the C4 position, methyl groups at the C10 and C13 positions, and an aliphatic side chain of 8 to 10 carbon atoms at C17 (Figure 1A). C4-SBIs display a variety of structural motifs: unsaturations at different positions of the B ring (Δ^7(8)^ or Δ^8(9)^ or Δ^9(11)^ or Δ^8,14^), a cyclopropanic cycle on the B ring, a methyl group at C14 and a methyl (or methylene) or ethyl (or ethylidene) at C24 on the side chain (Figure 1B). C4-SBIs are generally in low abundance contrasting with cellular amounts of pathway end-products. There are however organisms that contain substantial amounts of 4-methylsterols such as dinosterol implied in cold adaptation (Figure 1C) in dinoflagellates [26], or 4,4-dimethylsterols and 4-methyl-Δ^7^-sterols in the prokaryote *Methyloccocus capsulatus* [27]. Because dinosterol is restricted to dinoflagellates its sterane derivatives are used as biogeological markers of Phanerozoic sediments [28]. Alternatively, 4α-methyl-24-ethylcholestane may derive from C4-methylsterols from other yet unrecognized Proterozoic eukaryotic organisms that used C4-SBIs as membrane components (Figure 1C) [29]. The capacity of C4-SBIs like for instance cycloartenol to act as an efficient membrane structural component in primitive organisms has been discussed [30]. In this respect, a yeast mutant *erg7* deficient in lanosterol synthase (ergosterol-auxotrophic) could, however, live on C4-SBIs such as cycloartenol upon expression of a cycloartenol synthase [31,32].

C4-SBIs may be classified from an operational point of view according to amounts detected in an organism or tissue, as major C4-SBIs and transient C4-SBIs. Major C4-SBIs are present in few percents of total sterols, about several µg·g^−1^ dry weight, like for instance lanosterol in yeast, cycloartenol in *Artocarpus integrifolia* [33], or cycloeucalenol and obtusifoliol in plant tissues [34], and 24-ethylidenelophenol in *Hordeum vulgare* [35]. Transient C4-SBIs are intermediates of the sterol-C4-demethylation process catalyzed by a complex of enzymes (C4-DeMethylation Complex, C4DMC) and are generally not detected in sterol profiles under normal physiological conditions. These compounds are 4-hydroxymethylsterols, 4-formylsterols, 4-carboxysterols, canonical and non-canonical C4-SBIs and 3-ketosterols (Figure 2A).

The effectiveness of 4,4-dimethylsterols such as lanosterol (compared to cholesterol) in regulating membrane fluidity and supporting cellular functions in *Mycobacterium capricolum* was assessed by measuring microviscosity of membranes and establishing their capacity to promote prototrophic growth. Membranes of *M. capricolum* grown on medium containing 4,4-dimethylsterols or 4-methylsterols have microviscosity values found in between those of lanosterol (low value) and cholesterol (high value). These experiments demonstrated that the successive carbon removals at C14 of lanosterol then at C4 of 4,4-dimethylzymosterol and 4-methylzymosterol *en route* to cholesterol biosynthesis (Figure 2) progressively shaped a sterol molecule in order to sustain optimal cell growth [36]. This is in agreement with the identification of 4-methysterols in ancestral organisms [29,37,38].

Physiological roles of C4-SBIs have been described. Lanosterol in the brain is associated with a neuroprotective effect in Parkinson’s disease [39]. An increase of oligodendrocyte formation and remyelination was observed in the presence of C4-SBIs [40]. In mammal reproductive biology, Meiosis Activating Sterols (MAS) are major C4-SBIs found in follicular fluid (FF-MAS) and testicular tissue (T-MAS) (Figure 1B) [41,42,43]. FF-MAS are crucial for proper meiosis and for oocyte maturation in vitro [43,44]. Sterol biosynthetic flux analyzed in mice revealed a high rate of FF-MAS and T-MAS synthesis that defines cell-type specific pathways and also raised new hypothesis about the fate of T-MAS in testes (forming zymosterol, another sterol, a steroid hormone, or an excreted product) [45]. Synthetic FF-MAS and T-MAS were developed for further biological studies [46,47]. Human genetic diseases known as sterolosis are characterized by a dramatic accumulation of sterol intermediates including the immediate cholesterol precursors lathosterol and desmosterol (their accumulation causing lathosterolosis and desmosterolosis, respectively) but also of C4-SBIs causing severe alterations in development at early (embryo malformation) or later stages (skin anatomical changes) [48,49,50]. In *Caenorhabditis elegans*, 4-methylsterols are generated from cholesterol by an unusual C4-methylation enzyme that is only found in worms (Figure 3) [51]. In plants and mammals, transient C4-SBIs bearing a 4-formyl or 4-carboxy group were functionally linked to critical biological processes: the accumulation of 4α-carboxy-4β-methyl-24-methylenecycloartanol (oxojessic acid, Figure 1B) was shown to hamper proper auxin signaling in the model plant *Arabidopsis thaliana* [52], and 4α-formyl-lanosterol (Figure 1B) was described as a physiological ligand of RORγ, a protein that regulates lymphoid cell development [25].

## 2. Some Crucial Milestones in Deciphering the Sterol-Demethylation Process and Functions of C4-SBIs in Mammals

In mammals, the first demethylation step occurs at C14 position (Figure 2). This is achieved by a lanosterol-14α-methyl-demethylase (CYP51) [10,11,12] which removes the 14α-methyl group as formic acid resulting in a Δ^8,14^-diene product (Appendix A) [53]. This reaction requires NADPH and generates a 14α-formyloxysterol reaction intermediate on which CYP51 acts as a lyase in cleaving the C-C bond (Figure 2 and Appendix A). The CYP51A1 gene was identified in human and characterized by heterologous expression in bacteria [54]. The subsequent Δ^8,14^-sterol-Δ^14^-reduction (Figure 2 and Appendix A) has been the focus of considerable research effort over the last decade. In human, two different genes encode products that bear sterol-14-reductase activity, namely, LBR and TM7SF2 genes [19,20]. The LBR protein is bifunctional; it has a lamin B receptor (LBR) and sterol-14-reductase domains and is mainly acting on the cholesterol biosynthetic flux. The TM7SF2 protein, although exhibiting sterol-14-reductase activity, has not a well-defined function in cholesterol biosynthesis. The intracellular localization of these two proteins is different: LBR is addressed to the nuclear envelope, it bears a chromatin-binding N-terminus; TM7SF2 resides in the endoplasmic reticulum membranes [55,56,57,58].

The removal of C4 methyl groups as carbon dioxide during the conversion of lanosterol into cholesterol was shown years ago, suggesting that the demethylation reaction implied an β keto acid intermediate [59]. Another experimental evidence was provided by Bloch and co-workers who showed that the aerobic incubation of labeled 4-hydroxy-methylene-cholest-7-en-3-one in a rat liver homogenate resulted in a marked release of carbon dioxide from the reaction medium [60,61]. The subsequent isolation of 3-keto and 4α-acid reaction products supported the proposed mechanistic hypothesis for the C4 demethylation reaction: in rat liver microsomes, the incubation of ^14^C-labeled 4,4-dimethyl-5α-cholest-7-en-3-ol in the absence of NADPH led to the production of labeled carbon dioxide and mono-methylated products 3-keto-5α-cholest-7-en-3-one and 4-methyl-5α-cholest-7-en-3-one [62]. Similarly, the incubation of a 4-methylsterol produced carbon dioxide and a demethylated ketone at the C4 position [63]. In the presence of NADPH, these ketones are reduced to the corresponding 3α-alcohol by a 3-ketosteroid reductase [64]. The early stages of cholesterol biosynthesis studies and especially the identification of associated enzyme activities raised the question of the formation of the C4-carboxyl group preceding the carbon-carbon cleavage and loss of carbon dioxide, this based on a partial purification of an NAD+ decarboxylase [65,66]. Gaylor and co-workers showed that the oxidation of the methyl group at C4 to the corresponding acid required molecular oxygen and NADH and was sensitive to cyanide [67,68]. Also, the inhibition of C4-demethylation by snake venom phospholipases suggested the involvement of an NADH-dependent cytochrome b5 reducing system [63]. Finally, in recent decades, the complete set of genes coding the enzymes implied in the sterol-C4-demethylation step of mammalian cholesterol biosynthesis was identified particularly in deciphering some human genetic diseases; enzymes were thereafter biochemically characterized in heterologous systems [50,69,70]. The non-enzymatic protein ERG28 necessary for the activity of the C4DMC was lastly identified in human based on its yeast orthologs [71].

Functional studies of C4-SBIs have underlined critical biological properties of lanosterol. Lanosterol and oxysterols affect human cataracts [72]. A functional screening of molecules that bind alpha-crystallins (cryAA and cryAB) in vitro and reversed their aggregation identified 5-cholesten-3β,25-diol as an active compound, based on improved lens transparency in cataract models [73]. In another study, the direct relationship between congenital cataracts and lanosterol was shown by the elucidation of two causal mutations in the gene encoding lanosterol synthase [74]. The role of lanosterol in arresting cataract development was furthermore ascertained by its positive effect on protein disaggregation and the increase of lens transparency, both in vitro and in vivo, in rabbit and dog [74]. Further studies provided additional evidence to establish lanosterol firmly as an anti-cataract drug [74,75,76,77]. Although the molecular mechanism is not described, Quinlan [72] et al. have suggested that C4-SBIs, like lanosterol, could interact with small heat shock proteins, which function as sterol sensors regulating cellular and developmental processes. Lanosterol also has a tremendous impact on innate immunity [78]. The activation of Toll-Like Receptor 4 (TLR4) in macrophages is responsible for the transcriptional repression of CYP51, resulting in the accumulation of lanosterol. Such an accumulation of lanosterol, by genetic or by chemical inhibition, has a regulatory action on the immune response, membrane fluidity, ROS production and potentialize phagocytosis [78]. Considering cellular sterol homeostasis, lanosterol and 24,25-dihydrolanosterol are known to interact with the Insig signaling pathway that promotes the degradation of HMGR, a key enzyme of the mevalonate pathway [79]. Lanosterol and 24,25-dihydrolanosterol may also act as an oxygen sensor: in hypoxic conditions, the C14 and C4 demethylations rate is reduced, and consequently promote HMGR degradation, lowering thus the cholesterol biosynthetic flux [80].

The critical importance of C4-SBIs that are the reaction products of LBR and TM7SF2, two proteins bearing sterol-14-reductase domains, has emerged recently [55,56,58]. LBR and TM7SF2 act as regulators of TNFα expression in human, and skin papilloma development in mice [58,81,82,83,84,85]. The Greenberg skeletal dysplasia, the Renolds syndrome and Pelger–Huët anomaly are severe genetic diseases due to mutations in the LBR gene, causing a reduction in sterol-14-reductase activity and therefore promoting the accumulation FF-MAS, the substrate of the enzyme [86,87,88,89]. The molecular mechanism that is most probably at play in these diseases may be very close to an enhanced lipogenesis and the inhibition of cell proliferation mediated by the liver X receptor alpha (LXRα), to which binds the C4-SBI molecule FF-MAS [57]. Interestingly, a BODIPY-FF-MAS molecular probe was localized in nuclear lipid droplets of HepG2 cells. Such localization of FF-MAS is in line with the proposed regulatory role [57].

Functional genomics targeting components of the C4DMC led to highlights in human cholesterol biology. In a cancer cell line, the increased sensitivity to antagonists of an oncogenic epidermal growth factor receptor was revealed upon siRNA-based inactivation of SC4MOL and NSDHL leading to 4,4-dimethylzymosterol, 4-methylzymosterol, or 4-carboxysterol accumulation [90]. The inhibition of CYP51A1 suppressed the accumulation of these C4-SBIs and reversed the EGFR inhibitor sensibilization, rescuing cancer cell viability and EGFR degradation [90]. In human development, a hypomorphic temperature-sensitive allele of NSDHL causing the overaccumulation of 4-methylsterols in the cerebrospinal fluid was the cause of brain malformations typical of the CK syndrome (CKS) [49]. The SC4MOL-deficiency is an autosomal recessive lesion causing psoriasiform dermatitis, arthralgias, congenital cataracts, microcephaly, and developmental delay. Plasma sterol analysis showed a different cholesterol content in healthy individuals (140–176 mg·dL^−1^) versus patients (85–93 mg·dL^−1^). Most importantly, a ten-fold increase was obtained when measuring 4-methylsterols: 41–42 mg·mL^−1^ in patients plasma compared to 2.8–3.2 mg·mL^−1^ in healthy individuals [50]. In total, C4-SBIs presented a huge 500-fold increase in diseased individuals compared to healthy ones. No 4-carboxylmethylsterols neither 4-methylsterones were however detected. In such patients, fibroblasts had a 3-fold reduced rate of cell division, and immunocytes were abnormal, this was mimicked by applying aminotriazole, an inhibitor of SC4MOL/SMO. A causal relationship between the accumulation of C4-SBIs and skin barrier function, cell proliferation and immune regulation was then established [50]. Furthermore, the same authors demonstrated that C4-SBIs negatively regulate the epidermal growth factor receptor (EGFR), signaling and vesicular trafficking [91].

The Congenital Hemidysplasia with Ichthyosiform nevus and Limb Defects (CHILD) syndrome is a rare X-linked dominant disease with lethality for male embryos, sensorineural hearing loss, normal intelligence in females and one-sided cerebral hypoplasia [48]. More than 20 different alleles of the NSDHL gene were described [48,92,93]. 

Sterol analysis were performed in *nsdhl* mice: skin fibroblasts of bare patches of such mice contained about 20% of C4-SBIs in total sterols (71.4% of cholesterol, 18.2% of 4-methylsterols and 1.1 of 4,4-methylsterols), while control male mice had less than 0.1% of C4-SBIs and 99.9% of cholesterol [70]. The CKS consists of mild to severe intellectual disability in males, microcephaly, CNS malformation, seizures, hypotonia, dysphasia/speech delay, behavioral problems and possible psychopathological issues in female carriers. The CKS is lethal in females (whereas CHILD is lethal to males). Cerebrospinal fluid from CKS patients is enriched in 4-methylsterols and is low in cholesterol. It is also reported that CKS patients display a deficient hedgehog signaling [49]. No mutation (and associated human genetic disease) was reported in the case of C4D and ERG28. In mouse, the Rudolph mutant carries an allele of the C4D/HSD17B17 gene causing defective growth and patterning of the CNS, skeleton malformation, and an altered hedgehog signaling associated to an accumulation of zymosterone and 4-methylzymosterone [94]. The study of a conditional *nsdhl* mouse allele enabled a refined understanding of the link between cholesterol homeostasis and CNS at various developmental stages of pups. NSDHL deficiency and its associated accumulation of 4-methylsterols was responsible for defects in the cerebellum, hippocampus, cerebral cortex and led to early postnatal lethal phenotype [95]. At the cellular level, these defects were a thinner layer of granule cell precursors, which play a critical role in cerebral, cortical and hippocampal neuronal proliferation, differentiation and migration before birth. Using this *nsdhl* mouse line, an in vitro cell system was established from granule cell precursors to test the effect of 4-methylsterols on sonic hedgehog signaling (SHH). The obtained cell lines were cultivated with LDL supplementation and also ketoconazole treatment, in order to restore a cholesterol content, and to block the accumulation of 4-methylsterols, respectively. A hampered SHH signaling was correlated with the accumulation of 4-methylsterols. T-MAS (a functional 4-methylsterol), when added to wild-type cells obtained from granule precursors, mimicked perfectly the biogenetic accumulation otherwise noticed in conditional nsdhl cells, however no effect on SHH signaling was observed, most probably due to a mislocalization of T-MAS, or to the lack of bioconversion of T-MAS into an active yet unknown sterol-derived inhibitor of the SHH pathway.

C4-SBIs were described as essential players in the immune system. The binding capacity of C4-SBIs to the nuclear hormone receptor RORγt, an active component of lymphoid cells in thymus, was tested in vitro and in vivo [25]. 4-methylsterol biosynthetic intermediates in between the lanosterol to 4α-methylcholesta-8,24-dien-3-one (the substrate of C4D/HSD17B7) segment (Figure 2) exhibited the properties of ligands of RORγt albeit with significant affinity variations. 4-Methylsterols displayed the weaker affinity while oxygenated C4-SBIs like 4α-carboxy-4β-methylzymosterol (Figure 1) had a higher affinity. This study highlighted the regulatory role of bona-fide cholesterol biosynthetic intermediates upon immune system development and lymphoid functions. C4-SBIs have also a positive influence on mice oligodendrocyte formation and remyelination, as shown using sterol biosynthesis inhibitors. Inhibitors of C4-demethylation and of C14-reduction and Δ^8^-Δ^7^ isomerization (that promote the accumulation of C4-SBIs indirectly) led to the inactivation of a transcriptional program via the SREBP nuclear hormone receptors [40,96]. Further studies are required to identify firmly which C4-SBIs activate the SREBP machinery.

## 3. *Saccharomyces cerevisiae*, a Versatile Model for Sterol Genetics and Auxotrophy Studies

The yeast *S. cerevisiae* has established itself as a privileged model for the identification of sterol biosynthesis genes [98,99]. The advantages of yeast are plentiful: a sterol biosynthesis pathway similar to that of animals or plants enabling metabolic interferences, the possibility of homologous recombination to create loss-of-function mutants, its ability to have an uptake of exogenous sterols, to mention a few. The identification of the yeast SMO gene was published independently in 1996 by two teams. The Kaplan team screened a yeast mutant deficient in SMO activity based on its limited heme biosynthetic capacities [100]. The Bard team isolated the *erg25*/*smo* mutant by screening for SMO activity deficiency and identified the ERG25 gene (Figure 2) [101]. The yeast C4D was identified based on its functional homology with an NAD(P)-dependent cholesterol dehydrogenase gene of Nocardia sp. [102]; it complemented a corresponding deficient yeast (*erg26*) and *Candida albicans* mutants [103,104]. The yeast SKR gene encoding ERG27/SKR complemented a null mutant *erg27* deficient in 3-ketosteroid reductase (Figure 2) [105]. Gene expression analysis pointed out ERG28 and ergosterol biosynthetic genes within the same levels of expression [106]. The disruption of ERG28 induced a loss of C4-demethylation activity [107]. Protein interaction studies showed that ERG25, ERG26, ERG27, and ERG28 proteins are assembled in a complex tethered by ERG28 [108]. Although ergosterol biosynthesis was tremendously studied, some components of the machinery like ERG29 (an ER-associated protein) were unveiled just very recently [109].

The yeast *erg25* mutant contains high amounts of 4,4-dimethylsterols that are more effective than 4,4,14-trimethylsterols (like lanosterol) to disrupt growth. The lethality of *erg25* was overcome by mutations in ERG11 (lanosterol-14-demethylase) and SLU (suppressor of lanosterol utilization) to prevent the accumulation of 4,4-dimethylsterols and consequently ergosterol auxotrophy [110]. In the fission yeast *Schizosaccharomyces pombe*, the overexpression of ERG25 affected proper cytokinesis: the accumulation of 4,4-dimethylzymosterol-downstream products and further compositional changes in sterol/lipid-rich membrane domains led to defects in actomyosin ring positioning and maintenance [111]. The isolation of a yeast thermosensitive mutant *erg26-1* defective in the decarboxylation of 4-carboxy-4-methylsterols revealed the inefficiency of these C4-SBIs to support growth as bulk components. Protein-protein interaction studies pointed out a function for ERG26 in ERG7 regulation [112] and also in lipid homeostasis [107,112,113]. ERG29 was identified as an interactant or modulator of SMO/ERG25. The loss of ERG29 resulted in the accumulation of C4-SBIs and affected cell viability. In these yeast cells, an increase of mitochondrial oxidants and the degradation of the mammalian frataxin ortholog involved in mitochondrial iron-sulfur (Fe-S) cluster synthesis showed a link between sterol composition and iron metabolism in the mitochondrial compartment [109]. The expression of a gene cluster for helvolic acid production into *Aspergillus oryzae* NSAR1 has revealed the identification of C4-SBIs bearing anti-*Staphylococcus aureus* properties and unsual C4-demethylation enzymes [114]. In *S. pombe*, C4-SBIs have been identified as signaling molecules acting as oxygen sensor by interacting with the SRE1/SCP1 complex, which is equivalent to the mammalian SREBP regulatory pathway responsible for cholesterol homeostasis. Under conditions of low oxygen and cell stress, C4-SBIs accumulate and activate the transcription factor SRE1 [115].

## 4. The Plant-Specific Sterol-C4-Demethylation Process and Its Influence upon Development

Sterol-4α-methyl oxidase (SMO) is the enzyme of the C4DMC that acts first in the sequence of reactions. SMO enzymatic activities were initially studied with microsomal fractions of *Zea mays* coleoptiles incubated with radioactively labeled sterol substrates. This led to the clear-cut identification and characterization of two distinct SMO activities. These two SMO activities were shown to occur in a non-consecutive manner in the sterol pathway: a first SMO oxidizes 4,4,14-trimethylcyclopropylsterols such as 24-methylenecycloartanol, and a second SMO oxidizes 4α-methyl-Δ^7^-sterols such as 24-ethylidene lophenol (Figure 2 and Appendix A) [21]. It was also demonstrated that electrons are supplied via NADH to the oxygenase by the cytochrome b5/cytochrome b5 reductase system [116]. These distinct subcellular SMO activities corresponded in planta to the expression of distinct plant orthologs of the yeast SMO belonging to the SMO1 and SMO2 gene families [117]. When expressed in a yeast *erg25* mutant, SMO1 and SMO2 conferred different sterol biosynthetic capacity to their host [117,118]. Recently, the characterization of a cholesterol-specific biosynthetic segment in the Solanaceae (containing solanine or tomatine, which are steroidal glycoalkaloid derived from cholesterol) unveiled the function of additional SMO1 and SMO2 orthologs (named SMO3 and SMO4) that act specifically on C4-SBIs bearing cholesterol-type side chains [119]. Consequently, each SMO1 or SMO2 define distinct C4DMC comprising C4D, SKR, and ERG28. Two C4D (3β-hydroxysteroid dehydrogenase/C4-decarboxylase) were found to act redundantly in both types of C4DMC [120]. The plant SKR and ERG28 genes were functionally identified in protein-protein interaction assays and planta with the implementation of RNA silencing (RNAi) or knock-out T-DNA insertion lines [52]. Interestingly, the Arabidopsis SMO1-1 and SMO1-2 isoforms were identified as interactants of an Acyl-CoA-Binding Protein 1 (ACBP1) in yeast double hybrid assays, strongly suggesting a role for a SMO/ACBP1 complex in the regulation of lipid metabolism and particularly the activity of acyltransferases governing the production of triacylglycerols and sterol esters [121,122]. Also, it is proposed that the SMO1/ACBP1 complex controls plant development via an unknown lipid ligand that activates transcription factors like GLABRA2, HDG5, HDG10 [121,122]. The regulatory action of SMO1, possibly as a limiting step in phytosterol biosynthesis or by an unknown signaling activity of C4-SBIs, was illustrated in *A. thaliana* expressing jointly 3-hydroxy-3-methylglutaryl-coenzyme A reductase (HMGR) and SMO1 increasing by 54% in biomass [123].

The physiological functions of C4-SBIs were investigated using the elegant virus-induced gene silencing strategy (VIGS) in *Nicotiana benthamiana*, as already established in the case of the C14-demethylation step (CYP51; [124]). VIGS of SMO1 and SMO2 indicated deficiencies in distinct entities based on distinct sterol profiles: SMO1-silenced plants exhibited 4,4-dimethyl-9β,19-cyclopropylsterols as major sterols whereas SMO2-silenced plants had 4α-methyl-Δ^7^-sterols [117]. The same approach was successfully implemented to characterize a C4D gene in *N. benthamiana*: the dramatic reduction in gene expression resulted in the accumulation of the 4-carboxymethyl-4-methylsterol substrate of C4D in silenced leaves [120]. In SMO2 silenced plants, changes in the activity of the C4DMC caused a subsequent increase (compared to wild-type) of the ratio of C24-methylsterols to C24-ethylsterols in the sterol profiles. 24-Methylenelophenol is the substrate of SMO2 and also of the sterol-C24-methyltransferase SMT2 [125,126,127], and consequently defines a branching point in plant sterol biosynthesis (Figure 2). Therefore, the down-regulation or overexpression of SMO2 indirectly modulate the ratio of 24-methylsterols to 24-ethylsterols (mainly, campesterol to sitosterol), causing deleterious effects on growth [24]. Biotic interactions at the sterol metabolism interface were also studied in the context of silenced SMO genes in *N. benthamiana* to investigate the replication of tombusviruses (TBSV, tomato bushy stunt virus), a group of viruses depending on cellular membranes for replication. The authors also implemented a chemical treatment of plants with 6-amino-2-*n*-pentylthiobenzothiazole (APB), an inhibitor of the fungal SMO [128]. Silencing of SMOs and APB treatment reduced virus replication. Notably, APB was effective in slowing down virus replication in *N. benthamiana* protoplasts. Exogenous addition of campesterol and sitosterol in the medium rescued replication of the virus. The authors have also tested the effect of sterol biosynthesis inhibition by APB on tobacco mosaic virus (TMV) replication and showed that TMV accumulation was sterol-independent. The authors proposed two explanations accounting for the difference in replication between the TBSV and TMV in their host plant: i) tombusviruses proteins are integrated into membranes and interact with sterols; ii) each type of virus replicate in distinct subcellular compartments having specific sterol composition [128].

Functional aspects of C4-SBIs were investigated in *A. thaliana* by overexpressing or knocking-out genes of interest and scrutinizing their associated phenotype. *A. thaliana* overexpressing C4D displayed a short internode phenotype that was not rescued by brassinosteroids. The authors suggested that the accumulation of 3-ketosterols, the products of C4D like 22-hydroxy-5β-ergostan-3-one would alter membrane properties, auxin transporter activity and consequently growth and development (Figure 2) [129]. This conclusion is also in line with possible modification of the sterol composition of membrane microdomains, which are tremendously important in cellular homeostasis and signaling [130]. The characterization of loss-of-function *smo2* alleles in *A. thaliana* required double null mutants of both SMO2-1 and SMO2-2, to deal with genetic redundancy [117]. The complete loss of SMO2 was lethal or at least resulted in an early arrest in embryogenesis [131]. However, heterozygote (*smo2-1/smo2-1*, *smo2-2/+*) were dwarfs and late-flowering plants, with phenotypic features like small round dark green leaves reminiscent of some other sterol biosynthetic mutants bearing genetic defects in the conversion of Δ^7^-sterol intermediates to Δ^5^-sterols (campesterol and sitosterol) [132,133,134,135,136]. The sterol profiles of heterozygote *smo2-1*/*smo2-2* lines showed a marked accumulation of C4-SBIs such as 24-ethylidenelophenol up to 20% of the total and a decrease in campesterol and stigmasterol [131]. A careful examination of the phenotypic traits of *smo2* plants pointed out very clearly their impaired response to auxin [131]. The exogenous application of auxin or the introgression of *smo2* mutations in auxin overproducer lines such as those overproducing free IAA upon enhancement of the YUCCA gene expression resulted in the rescue a wild-type developmental phenotype in *smo2* loss-of-function mutants [131]. It is conceivable that the accumulation of C4-SBIs alters plasma membrane properties, particularly the proper localization of auxin efflux PIN proteins, as shown earlier [137,138]. Alternatively, C4-SBIs may act as components of auxin signaling. This was proposed by independent studies consisting in altering the expression of an enzyme of C4DMC (C4D; [129]) or of ERG28, the non-enzymatic protein that tethers the C4DMC [52]. In the latter study, several *erg28* knocked-down Arabidopsis lines displayed an abnormal accumulation of the transient C4-SBI oxojessic acid (in µg·g^−1^ fresh weight amount compared to undetectable signals in wild-type plants). Phenotypes of such plants were reminiscent of an auxin disrupted homeostasis: in fact, experimental evidence supports the function of oxojessic acid as an inhibitor of polar auxin transport [52]. Taken together these results point out a novel critical role for C4-SBIs on growth and development that is distinct from the status of sterol end-products or brassinosteroids.

## 5. Caenorhabditis elegans: A Sterol Auxotroph with an Extraordinary C4-Methylation Capacity

Nematodes are sterol auxotrophs, just like insects and some other invertebrates. These organisms live on exogenous sterols provided by their diet. They also convert a proportion of cholesterol into steroid hormones known as the dafachronic acids, which bind the nuclear hormone receptor DAF12 responsible for reproductive development (Figure 3) [139]. In fact, the biogenesis of these compounds requires C4-desmethylsterols as substrates (i.e., cholesterol, with a free C4 position) and the action of a 3-hydroxysteroid dehydrogenase/Δ^5^/Δ^4^ isomerase (HSD-1) for the conversion of cholesterol to cholest-4-en-3-one en route to dafachronic acid (Figure 3) [51,139]. The arrest of the reproductive cycle upon environmental stress requires the inactivation of dafachronic acid biogenesis that enables unbound DAF12-mediated larval entry into the dauer stage, a particular diapause. Quite uncommon in the eukaryotic tree of life, a sterol-C4-methyltransferase named STRM-1 catalyzes the addition of a single methyl group provided by *S*-adenosyl-methionine onto the sterol tetracyclic moiety. The products of the methylation reaction like lophenol or its isomeric 4α-methyl-5α-cholest-8(14)-en-3β-ol are sterol biosynthesis end-products rather than C4-SBIs in this particular context (Figure 3) [51]. The enzymatic reaction catalyzed by STRM-1 has not been investigated into much detail. This sterol methylation restricted to nematodes regulate the biologically active amounts of dafachronic acids, pointing out the tremendous importance of 4-methylsterols in development since it is the C4-methylated product that triggers the entry of the worm into the dauer stage [139,140].

## 6. Bacteria Evolved Their Specific C4-Demethylation Enzymes

The capacity to synthesize sterols is usually not a prokaryotic feature. However, genes encoding the steroidal triterpene forming enzyme 2,3-oxidosqualene cyclase (OSC) were found in 34 bacterial genomes from several phyla (myxobacteria, methylococcales, rhizobiales, planctomycetes, and some others), thus predicting putative or minimal sterol pathway comprising 2,3-oxidosqualene cyclization products and subsequent C14 and C4 demethylations of those (Figure 4) [141]. Interestingly, γ-proteobacterial aerobic methanotrophs like *Methylococcus capsulatus* are characterized by a C4 demethylation process removing one single methyl group at C4-position of 4,4-dimethylsterols, whereas δ-proteobacterial myxobacteria can remove both methyl groups at C4 like it is the case in eukaryotes [141]. The single C4 demethylation that is typical of *M. capsulatus* is catalyzed by the consecutive action of two enzymes (Figure 4 and Appendix A). These sterol demethylation (Sdm) enzymes are strikingly different from the eukaryotic C4-demethylation enzymes described above. SdmA is a Rieske-type oxygenase that catalyzes three successive oxidations of the C4β methyl group of 4,4-dimethylsterols, whereas the non-heme oxygenase SMO performs the successive oxidation reactions of the C4α methyl group of 4,4-dimethylsterols in eukaryotes. Rieske-type oxygenases have been described in sterol pathways of dafachronic acids in *C. elegans* [142] and of the protist *Tetrahymena thermophila* [143,144] where that type of enzymes acts as a cholesterol-7-desaturase. SdmB is the second enzyme responsible for both decarboxylation and ketoreduction steps. The reversibility of the last step has been discussed [141]. These findings demonstrate that a sterol-C4-demethylation process has evolved twice independently and that the bacterial Sdm enzymes are functionally restricted to demethylate at C4β without any further oxidation at C4α, explaining thus the production of C4-SBIs as pathway end-products in methanotrophs otherwise used as geological biomarkers. The function of 4-methylsterols in bacteria is not clearly understood. A role in adaptation to environmental constraints like water salinity or limitation in oxygen has been suggested [141].

## 7. Inhibitors of C4-SBIs Accumulation In Vivo, Canonical and Non-Canonical C4-SBIs, and Conjugated forms

The overall chemical or genetic inhibition studies of C4-demethylation steps of cholesterol (plants and mammals), ergosterol (fungi, algae) or phytosterol biosynthesis demonstrate that the removal of both methyl groups at C4-position are necessary for proper growth or development. The accumulation of lanosterol results from the inhibition of the sterol-C14-demethylase, that is a P450-dependent mono-oxygenase. The class of ‘azoles’, that includes imidazoles and triazoles, is widely used as therapeutic and agricultural antifungal drugs [145,146]. For instance, clotrimazole is used to monitor C4-SBI accumulation in yeast and human (Figure 5) [40,147]. It is worth noting that the accumulation of C4-SBIs may be caused by inhibitors acting in fact on the sterol-C14-reduction and sterol-C8-isomerization steps like the morpholine derivative amorolfine or the compound AY9944 [40,57,148].

It is therefore relevant to envision the SMO, C4D, and SKR enzymes as interesting target sites for new fungicides or herbicides. In yeast, APB (Figure 5) inhibited sterol-C4-demethylation [149]. APB was assayed in vitro on maize coleoptile microsomal SMO1 and SMO2 enzymatic activities: surprisingly, APB did not affect SMO1 whereas it displayed a limited inhibition of SMO2 (compared to the strong effect observed on the yeast SMO) [150,151,152]. Other compounds like PF1163A and PF1163B were isolated from *Penicillium* sp. PF1163A (Figure 5) caused a steady accumulation of 4,4-dimethylzymosterol in yeast indicating SMO as the target of these new antifungal antibiotics [153,154,155]. Garlic extract and 17-hydroxyprogesterone inhibited human SMO [57,156,157], 3-amino-1,2,3-triazole (ATZ) was described as a potent SMO inhibitor in mice [91,158,159], the cholesterol-lowering oxysteroid FR171456 (Figure 5) was recently characterized for its inhibitory property on C4-decarboxylation enzymes (NSDHL in human, ERG26 in yeast) [160]. Fenhexamid (a hydroxyanilide) and fenpyrazamine (an aminopyrazolinone) are antifungal agents presently on the market. The fungal sterol profiles established in the presence of fenhexamid displayed an accumulation of 3-ketosterols (zymosterone), showing that the inhibition of SKR was most probably the reason of fungitoxicity [161].

The classification of canonical and non-canonical C4-SBIs was proposed by the WD Nes (Texas Tech University, Lubbock, TX, USA) and the Littman (Howard Hugues Medical Institute, Chevy Chase, MD, USA) groups [25]. This definition is based on a thorough chemical analysis of oxysterols binding the nuclear receptor RORγ, this in several genetic backgrounds of mice carrying loss-of-function mutations of the enzymes CYP51 (sterol-C14-demethylation) or in SC4MOL/SMO (sterol-C4-demethylation). Canonical oxygenated metabolites derived from the major C4-SBIs T-MAS and 4α-methylzymosterol (Figure 2A) are generated by three successive SMO-catalyzed oxidations of the methyl group at C4 yielding a 4-hydroxymethylsterol, a 4-formylsterol and a 4-carboxysterol transient sterol biosynthesis intermediates (Figure 2A). These C4-SBIs are usually not detected in routine sterol profiling of given organs or tissues and for this reason, could be even considered as cryptic. However, an inhibitor of SMO fed to a yeast microsomal fraction enabled the identification of 4-hydroxymethylsterols, namely, 4β-methyl-4α-hydroxymethyl-5α-cholesta-8,24-dien-3β-ol and 4α-hydroxymethyl-5α-cholesta-8,24-dien-3β-ol [152]. Likewise, carboxysterols and ketosterols were identified in yeast, plants, and mammals following chemical or genetic inhibition. In the case of plants, C4D gene silencing in *N. benthamiana* led to a remarkable accumulation of 3β-hydroxy-4β,14-dimethyl-5α-ergosta-9β,19-cyclo-24(28)-en-4α-carboxylic acid [120].

In mammals, non-canonical oxygenated C4-SBIs are produced by the action of SMO on lanosterol before its demethylation at C14 (Figure 2A). Lanosterol was shown to be a substrate of the *S. cerevisiae* SMO [152]. Non-canonical oxygenated compounds are therefore 4-hydroxymethyl-14-methylsterols, 4-formyl-14-methylsterols, and 4-carboxymethyl-14-methylsterols [25]. Mice thymus contained concentrations of about 60 nM 4-hydroxymethyl-4,14-dimethylcholesta-8,24-dien-3β-ol [25]. Non-Canonical C4-SBIs bearing a 14-hydroxymethyl or 14-carboxymethyl group were identified in previous studies on the C14-demethylation reaction. 14-hydroxymethyl-4,4-dimethylcholesta-8,24-dien-3β-ol accounted for about 1% of total cellular sterol in hepatocytes [162]. The range of non-canonical C4-SBIs is therefore due to the versatility of SMOs that can react as 4α-methylsterol-oxidases on a variety of 4,4-dimethyl- and 4-methylsterol substrates [152].

Conjugated forms of C4-SBIs have been over-looked in biology. Many of these compounds belong to the so-called specialized metabolites (of plants, of protists, of bacteria). Lanosterol glycosides were reported in *Muscari paradoxum* [163]. Cycloartenol esters of fatty acids were found in *Ixora coccinea* [164]. In the marine diatom *Skeletonema marinoi*, sterol sulfates were associated with programmed cell death that occurs as a mechanism regulating phytoplankton blooms [165]. In mammals, the inhibition of SMO by an aminotriazole drug fed to rats resulted in a peroxisomal accumulation of 4α-methylcholest-7-en-3β-ol and its corresponding ester of fatty acids (18% of esters and 82 % of free 3β-OH form). In the same tissues, 4,4-dimethylcholest-8-en-3β-ol was found in its free form only, whereas the total cholesterol included 12% of cholesterol esters [158,159]. C4-SBIs were found as sulfates in patients suffering familial hypercholesterolemia and treated with partial ileal bypass surgery [166]. The function of these sulfates was not well perceived until recent studies in mice proposed for sterol sulfates the role of agonists of the endogenous retinoic acid receptor-related orphan receptor γ (RORγ). This receptor plays a crucial role in the differentiation of lymphocytes and autoimmune diseases [167]. The limited current understanding of the physiological role of conjugated C4-SBIs (glycosides, lipid esters, sulfolipids) as signaling molecules will require further research initiatives.

## 8. Concluding Remarks

Genes and their products responsible for sterol-C4-demethylation in mammals, yeast, plants, and bacteria have been quite well described by several groups over the last years, as shortly reviewed above. There are striking differences between bacteria and other organisms (protists, metazoans) regarding C4-demethylation mechanisms recruited during evolution. Plants use distinct C4DMC defined by substrate specificity: SMO1-based demethylation complex of 4,4-dimethylsterols and SMO2-based demethylation complex of 4-methylsterols, whereas other organisms demethylate 4,4-dimethylsterols and 4-methylsterols consecutively with a single SMO-based complex (of three enzymes and a tethering protein ERG28). In eukaryotes, genetic or chemical inhibition of the sterol-C4-demethylation may lead to the accumulation of significant amounts of C4-SBIs or transient C4-SBIs, but also of their oxygenated derivatives classified as canonical and non-canonical. The review of biological activities of 4-methylsterols characterized so far in different kingdoms shows clear common features. In yeast, in plants or mammals, the accumulation of C4-SBIs (including oxygenated derivatives) has deleterious effects on growth and development. In yeast and mammals, the role of C4-SBIs in cell division was shown. In plants and mammals, 4-methylsterols and 4-carboxysterols act as signaling molecules interfering with major pathways like auxin in plants and immune system in mammals. The major challenge remains the identification of physical interactions of sterol ligands with their targets. Another critical issue is the analytical scale of those biosynthetic intermediates: just like some oxysterols or brassinosteroids, 4-carboxysterols may be present at very low concentration, e.g., at “hormone-dose” and are therefore not detected in sterol profiles. For example in *A. thaliana*, bulk sterols were quantified 100–200 µg·g^−1^ fresh weight, canonical C4-SBIs 0.1–0.5 µg·g^−1^ fresh weight, but brassinolide 4 × 10^−5^ µg·g^−1^ fresh weight. Finally, the fate of C4-SBIs as metabolic products requires further investigations, regarding the enzymes implied in this process, and the type of formed products, like for instance hydroxysteroids or sulfates as shown in a study of RORγt receptors [167].

## Figures and Tables

**Figure 1 molecules-24-00451-f001:**
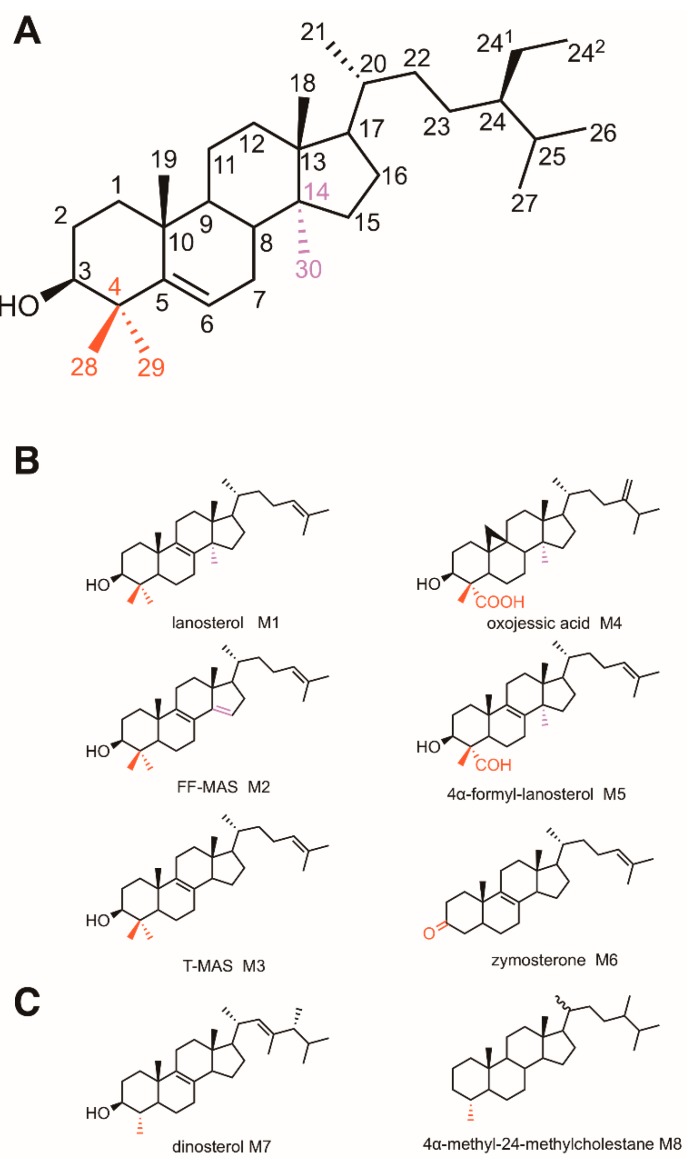
Sterol and 4-methylsterol structures. (**A**) carbon numbering. (**B**) some compounds described in this article. (**C**) dinosterol and a sterane, a biogeological marker.

**Figure 2 molecules-24-00451-f002:**
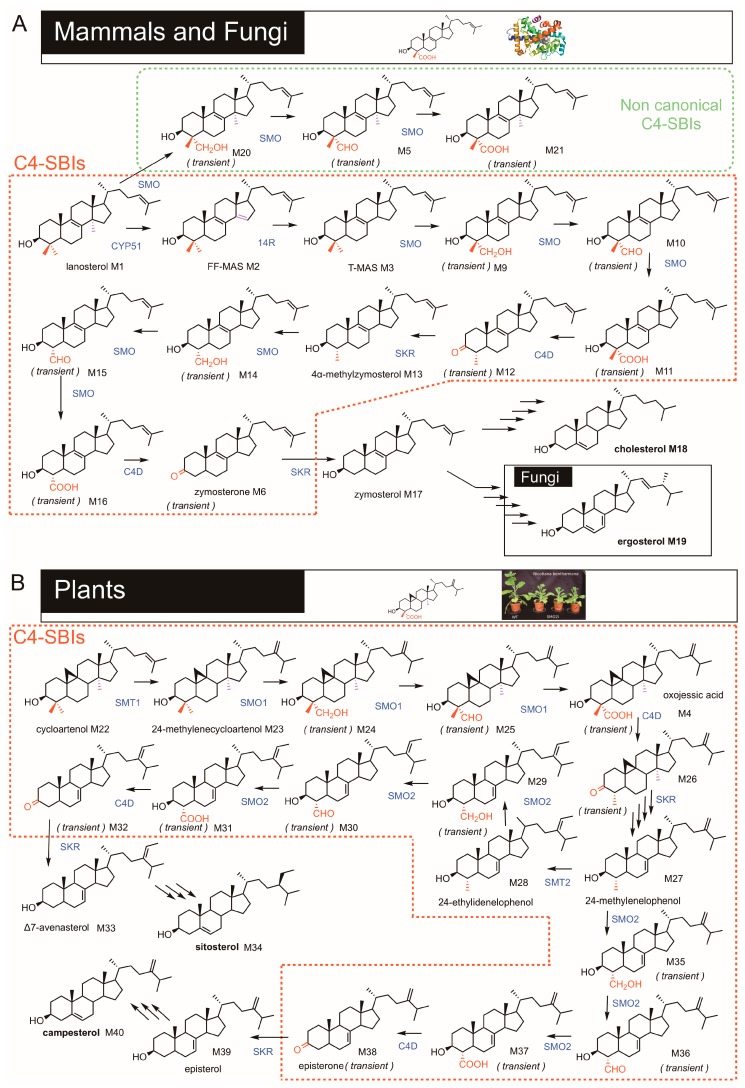
C4-demethylation pathways in mammals and fungi, and plants. Sterol nomenclature is given in Table 1. (**A**) pathways in mammals and fungi; (**B**) pathway in plants; C4-demethylation in eukaryotes: SMO, sterol-4α-methyl-oxidase; C4D, 3β-hydroxysteroid dehydrogenases/C-4 decarboxylase; SKR, sterone ketoreductase, C14-demethylation: CYP51, lanosterol-C14 demethylase. SMT, sterol methyltransferase; 14R, sterol-14-reductase. Each arrow represents an enzymatic step. Graphical insets are from references ([24] in top and [25] bottom panels).

**Figure 3 molecules-24-00451-f003:**
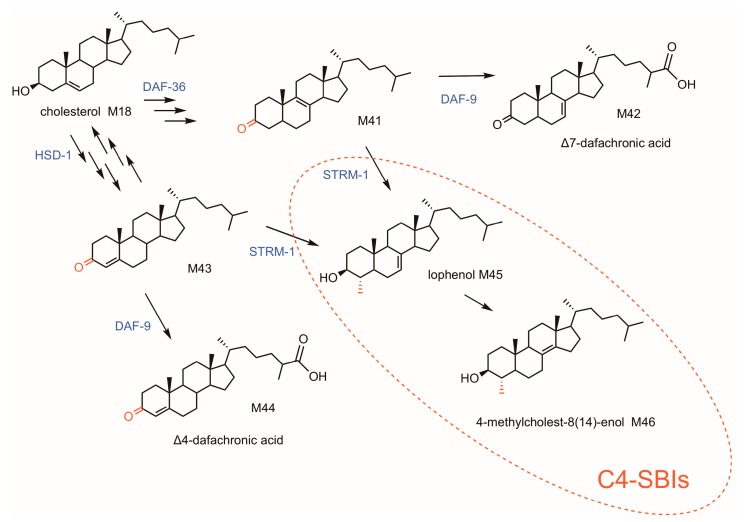
Dafachronic acid synthesis in *Caenorhabditis elegans*. Sterol nomenclature is given in Table 1. C4-demethylation in eukaryotes: HSD-1, 3-hydroxysteroid dehydrogenase/Δ5/Δ4 isomerase (HSD-1); STRM-1, Sterol 4-C-methyltransferase; DAF-9, steroid cytochrome P450 hydroxylase; DAF-36, cholesterol 7-desaturase. Each arrow represents an enzymatic step.

**Figure 4 molecules-24-00451-f004:**
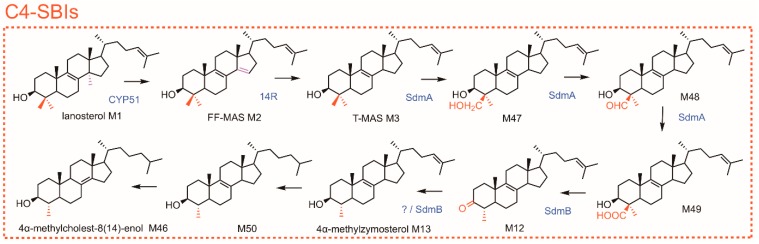
Sterol pathways in *Methylococcus capsulatus*. Sterol nomenclature is given in Table 1. CYP51, lanosterol-C14 demethylase; 14R, sterol-14-reductase; Sdm, sterol demethylase. Each arrow represents an enzymatic step.

**Figure 5 molecules-24-00451-f005:**
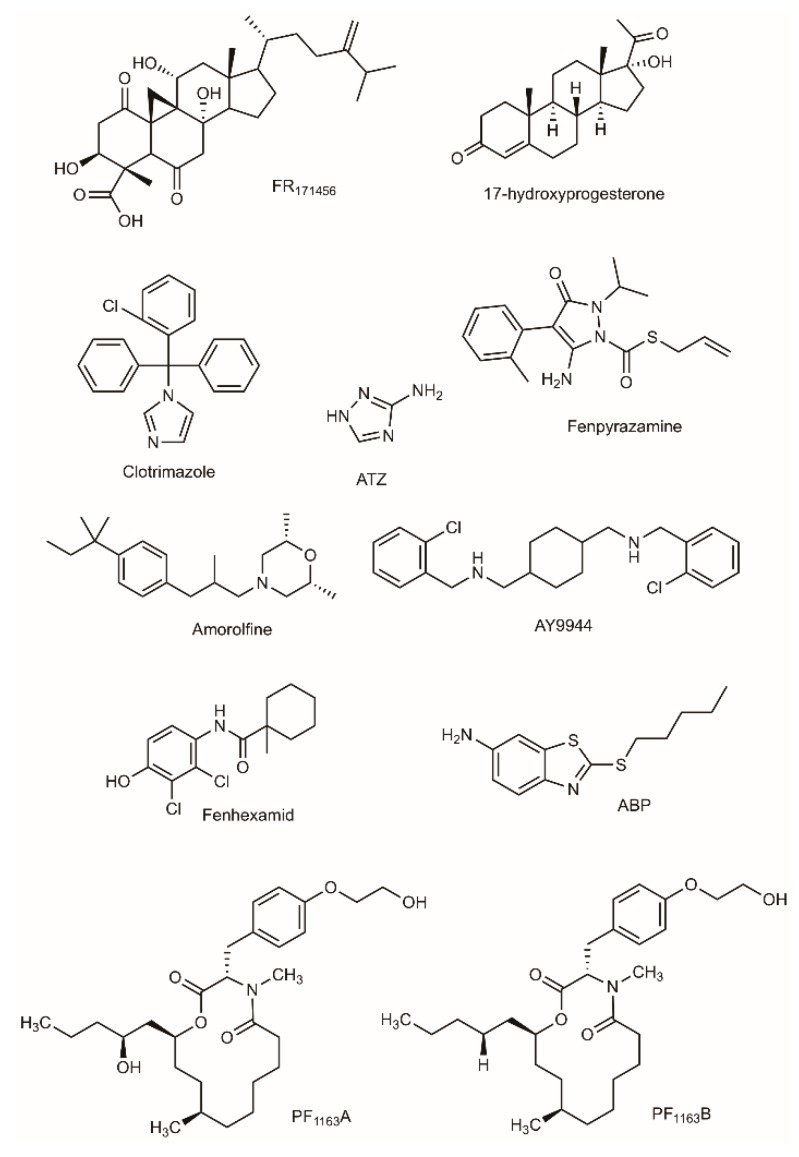
Chemical inhibitors for C4-SBI accumulation.

**Table 1 molecules-24-00451-t001:** IUPAC sterol nomenclature [97].

ID	Common Name	IUPAC
M1	lanosterol	lanosta-8,24-dien-3β-ol
M2	FF-MAS	4,4-dimethyl-5α-cholesta-8,14,24-trien-3β-ol
M3	T-MAS, 4,4-dimethylzymosterol	4,4-dimethyl-5α-cholesta-8,24-dien-3β-ol
M4	oxojessic acid, CMMC	4α-carboxy-4β,14α-dimethyl-9β,19-cyclo-5α-ergosta-24(24^1^)-en-3β-ol
M5	4α-formyl-lanosterol	4α-formyl-4β,14α-methyl-cholesta-8,24-dien-3β-ol
M6	zymosterone	5α-cholesta-8,24-dien-3-one
M7	dinosterol	4α,23,24-trimethyl-5α-cholesta-22-en-3β-ol
M8	4α-methyl-24-ethylcholestane	4α,24-methyl-cholestan-3β-ol
M9	4α-hydroxymethyl-4β-methyl-zymosterol	4α-hydroxymethyl-4β-methyl-cholesta-8,24-dien-3β-ol
M10	4α-formyl-4β-methylzymosterol	4α-formyl-4β-methyl-cholesta-8,24-dien-3β-ol
M11	4α-carboxy-4β-methylzymosterol	4α-carboxy-4β-methyl-cholesta-8,24-dien-3β-ol
M12	3-keto-4α-methylzymosterol	4α-methyl-5α-cholesta-8,24-dien-3-one
M13	4α-methylzymosterol	4α-methyl-5α-cholesta-8,24-dien-3β-ol
M14	4α-hydroxymethylzymosterol	4α-hydroxymethyl-5α-cholesta-8,24-dien-3β-ol
M15	4α-formylzymosterol	4α-formyl-5α-cholesta-8,24-dien-3β-ol
M16	4α-carboxyzymosterol	4α-carboxy-5α-cholesta-8,24-dien-3β-ol
M17	zymosterol	5α-cholesta-8,24-dien-3β-ol
M18	cholesterol	cholest-5-en-3β-ol
M19	ergosterol	ergosta-5,7,22E-trien-3β-ol
M20	-	4α-hydroxymethyl-4β,14α-methyl-cholesta-8,24-dien-3β-ol
M21	-	4α-carboxy-4β,14α-methyl-cholesta-8,24-dien-3β-ol
M22	cycloartenol	9β,19-cyclo-lanost-24-en-3β-ol
M23	24-methylenecycloartanol	24-methylene-9β,19-cyclo-lanost-3β-ol
M24	4-hydroxymethyl-24-methylenecycloartanol	4α-hydroxymethyl-24-methylene-9β,19-cyclo-lanost-3β-ol
M25	4-formyl-24-methylenecycloartanol	4α-formyl-24-methylene-9β,19-cyclo-lanost-3β-ol
M26	cycloeucalenone	24-methylene-9β,19-cyclo-lanost-3-one
M27	24-methylenelophenol	4α-methyl-24-methylene-cholest-7-en-3β-ol
M28	24-ethylidenelophenol	4α-methyl-24Z-ethylidene-cholest-7-en-3β-ol
M29	4-hydroxymethyl-24-ethylidenelophenol	4α-hydroxymethyl-24Z-ethylidene-cholest-7-en-3β-ol
M30	4-formyl-24-ethylidenelophenol	4α-formyl-24Z-ethylidene-cholest-7-en-3β-ol
M31	4-carboxy-24-ethylidenelophenol	4α-carboxy-24Z-ethylidene-cholest-7-en-3β-ol
M32	avenasterone	24Z-ethylidene-cholest-7-en-3-one
M33	Δ7-avenasterol	24Z-ethylidene-cholest-7-en-3β-ol
M34	sitosterol	stigmast-5-en-3β-ol
M35	4-hydroxymethyl-24-methylenelophenol	4α-hydroxy-24Z-methylene-cholest-7-en-3β-ol
M36	4-formyl-24-methylenelophenol	4α-formyl-24Z-methylene-cholest-7-en-3β-ol
M37	4-carboxy-24-methylenelophenol	4α-carboxy-24Z-methylene-cholest-7-en-3β-ol
M38	episterone	24-methylene-cholest-7-en-3β-one
M39	episterol	24Z-methylene-cholest-7-en-3β-ol
M40	campesterol	campest-5-en-3β-ol
M41	lathosterone	cholest-7-en-3-one
M42	Δ7-dafachronic acid	(25s)-3-oxocholest-7-en-26-oic acid
M43	-	cholest-4-en-3-one
M44	Δ4-dafachronic acid	(25s)-3-oxocholest-7-en-26-oic acid
M45	lophenol	4α-methyl-cholest-7-en-3β-ol
M46	4α-methylcholest-8(14)-enol	4α-methyl-5α-cholest-8(14)-en-3β-ol
M47	4β -hydroxymethyl-4α -methyl-zymosterol	4β -hydroxymethyl-4α -methyl-cholesta-8,24-dien-3β-ol
M48	4β -formyl-4α -methyl-zymosterol	4β -formyl-4α -methyl-cholesta-8,24-dien-3β-ol
M49	4β -carboxy-4α -methyl-zymosterol	4β -carboxy-4α -methyl-cholesta-8,24-dien-3β-ol
M50	-	4α-methyl-5α-cholesta-8-en-3β-ol

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
