# Peer review of "Metabolism and Biological Activities of 4-Methyl-Sterols"

_molecules, 2019, doi:10.3390/molecules24030451_

Round 1
Reviewer 1 Report
In this review Darnet and Schaller cover the biosynthesis and biological effects of C4-methylated sterols across all kingdoms. In looking for past related reviews, we found examples that tackled sterol synthesis more broadly, including in mammals or plants.Some reviews also tackled the biological effects of specific C4-methyl sterols (eg meiosis-activating sterols). But no concise review of the C4 methyl portion of sterol synthesis was found. While I think this topic is interesting and not well-reviewed elsewhere, issues of scope, clarity, and accuracy should be considered prior to publication.
It may make sense to restrict the scope more explicitly to the C4 demethylation complex, since C4 demethylation is the major thrust of the manuscript and recent references relating to the biological effects of C4 methyl sterols prior to C4 demethylation are missing (eg, roles for lanosterol in cataracts published in Science or in immune function in Cell Reports). In multiple instances CYP51 inhibitors like ketoconazole are stated to block the formation of C4 methyl sterols, but CYP51 inhibition just leads to elevated levels of a different C4 methyl sterol, lanosterol. Likewise, sterol 14 reductase is not substantially discussed, and recent studies showing that two enzymes can catalyze sterol 14 reduction in humans are not included. The review is much more comprehensive regarding the C4 demethylation enzymes and sterols.
A different scope question relates to the discussion of mammalian, plant, yeast, and bacterial enzymes for C4 demethylation together with the known biological effects of C4 sterols across the kingdoms. This is an ambitious scope, and I think it would be easier to readers to follow if the review followed a clearer outline. Defining all biosynthesis pathways for all kingdoms first, followed by a separate section relating to biological effects could be easier for readers to uptake.
Finally, I want to flag a number of issues relating to clarity and accuracy.
Fig 1b zymosterone has a C14 methyl group
Fig 2b end product should be FF-MAS, it’s missing a C14 olefin currently.
Aren’t Figure 2a and Figure 3 (mammalian section) identical? Why include twice?
Fig 4 amorolfine is labeled “TM7SF2” which is the gene name for an enzyme amorolfine inhibits. Same issue in the text.
Line 129: beta keto acid, not alpha keto acid
Line 405: I drew the opposite conclusion from this paper, that SREBP2 was unlikely to be involved.
A reference to consider adding: 17-hydroxy progesterone as an SC4MOL inhibitor (https://www.ncbi.nlm.nih.gov/pubmed/28499814),
Stylistic point: I would avoid use of Comic Sans (Fig 2).
Author Response
Reviewer 1
In this review Darnet and Schaller cover the biosynthesis and biological effects of C4-methylated sterols across all kingdoms. In looking for past related reviews, we found examples that tackled sterol synthesis more broadly, including in mammals or plants. Some reviews also tackled the biological effects of specific C4-methyl sterols (eg meiosis-activating sterols). But no concise review of the C4 methyl portion of sterol synthesis was found. While I think this topic is interesting and not well-reviewed elsewhere, issues of scope, clarity, and accuracy should be considered prior to publication.
It may make sense to restrict the scope more explicitly to the C4 demethylation complex, since C4 demethylation is the major thrust of the manuscript and recent references relating to the biological effects of C4 methyl sterols prior to C4 demethylation are missing (eg, roles for lanosterol in cataracts published in Science or in immune function in Cell Reports). In multiple instances CYP51 inhibitors like ketoconazole are stated to block the formation of C4 methyl sterols, but CYP51 inhibition just leads to elevated levels of a different C4 methyl sterol, lanosterol. Likewise, sterol 14 reductase is not substantially discussed, and recent studies showing that two enzymes can catalyze sterol 14 reduction in humans are not included. The review is much more comprehensive regarding the C4 demethylation enzymes and sterols.
We thank Reviewer#1 for valuable comments. As suggested, we have opened the scope of the review to the biological effects of C4 methyl sterols prior to C4 demethylation. Thus, aspects of lanosterol biology like its function in cataracts are described. Some additional relevant references on FF-MAS have been quoted as well. Furthermore, we have more precisely documented the C14 demethylation and C14-reduction steps in the section "An introduction to 4-methylsterols", which is now providing a better introduction to the limited scope of the C4 demethylation. Significant references about the LBR and TM7SF2 genes were added.
A different scope question relates to the discussion of the mammalian, plant, yeast, and bacterial enzymes for C4 demethylation together with the known biological effects of C4 sterols across the kingdoms. This is an ambitious scope, and I think it would be easier to readers to follow if the review followed a clearer outline. Defining all biosynthesis pathways for all kingdoms first, followed by a separate section relating to biological effects could be easier for readers to uptake.
We thank once again Reviewer#1 for this sound suggestion. We have modified the outline of the manuscript in order for the reader to progress easier through the text parts. The general biosynthesis pathways were described in a first section and then each kingdom in a separate section headed by an explicit title (sub-heading), as follows:
- An introduction to 4-methylsterols
- Some crucial milestones in deciphering the sterol-demethylation process and functions of C4-SBIs in mammals
- Saccharomyces cerevisiae, a versatile model for sterol genetics and auxotrophy studies
- The plant-specific sterol-C4-demethylation process and its influence upon the development
- Caenorhabditis elegans: a sterol auxotroph with extraordinary C4-demethylation capacity
- Bacteria evolved their C4-specific C4-demethylation enzymes
- Inhibitors of C4-SBIs accumulation in vivo, canonical and non-canonical C4-SBIs, and conjugated forms
Finally, I want to flag a number of issues relating to clarity and accuracy.
Fig 1b zymosterone has a C14 methyl group
We have corrected this mistake.
Fig 2b end product should be FF-MAS, it’s missing a C14 olefin currently.
We have corrected this mistake.
We apologize for the lousy quality of the previous drawings (14-double bond).
Aren’t Figure 2a and Figure 3 (mammalian section) identical? Why include twice?
We have now reorganized Figures. The former Figure 3, illustrating the C4 demethylation mechanism appears now as a supplementary Figure S1. We propose to the reader a Figure 2 showing Mammals and fungi, and Plants C4-demethylation pathways, a Figure 3 for cholesterol and dafachronic acid metabolism in C. elegans, and a Figure 4 showing bacterial C4-SBIs.
Fig 4 amorolfine is labeled “TM7SF2” which is the gene name for an enzyme amorolfine inhibits. Same issue in the text.
We have corrected this mistake (Figure and text).
Line 129: beta keto acid, not alpha keto acid
This mistake was corrected in the text.
Line 405: I drew the opposite conclusion from this paper, that SREBP2 was unlikely to be involved.
We have modified the text. A decrease of cholesterol or oxysterols has been reported as activating SREBP2. Consequently, the increase of C4-SBIs (oxysterols) should decrease the SREBP2 activation.
A reference to consider adding: 17-hydroxy progesterone as a SC4MOL inhibitor (https://www.ncbi.nlm.nih.gov/pubmed/28499814),
As suggested, we have included this inhibitor of SC4MOL in Figure 5, and also some other molecules, described as C14R and CYP51 inhibitors.
Reviewer 2 Report
Darnet and Schaller review the current knowledge on the functional roles of C4-methylsterols and properties of the pertinent biosynthetic enzymes. In general, this is a very comprehensive overview which provides readers a clear comparison and contrast between C4-methylsterol functions in mammals, fungi, plants and bacteria. Given the lack of recent reviews on the similar topic, this work is considered a significant contribution to the field. Some minor comments are listed as follows:
1) Line 11, the first letter should be capitalized (i.e. 4,4-Dimethylsterols). There are many other instances throughout the text and Table 1.
2) Line 13, “biochemical inhibitions” should be “biochemical inhibition”.
3) Line 33, please add “in plants” after “phytosterols”.
4) Line 46, “organism” should be “organisms”.
5) In Figures 1 and 4, please provide the full names besides abbreviations wherever possible.
6) Line 53, “Sterol and 4-Methylsterol” should be “Sterol and 4-methylsterol”.
7) Line 68, “Methylococcus capsulatus” should be italicized.
8) Line 92, “accumulations” should be “accumulation”.
9) In Figure 3, it is not clear how the plant photo in the graphical inset is related to auxin signaling without labeling.
10) Line 107, “… from references [91; top panel] and [37; bottom panel]” should be “… from references [91 in top panel and 37 in bottom panel]”.
11) Line 113-114, “A difference between sterol pathways of mammals and fungi, and plants is …” should be “A difference between sterol pathways of mammals and fungi versus plants is …”.
12) Line 141, “C4-carboxy group” should be “C4-carboxyl group”.
13) In Table 1, the numerous molecules are categorized into M1 to M9. The IDs should be explained.
14) Line 159, “uptake” should not be used as a verb.
15) Line 164, “a NAD(P)-dependent” should be “an NAD(P)-dependent”.
16) Line 175, please delete “this”.
17) Line 185, please add reference after “cell viability”.
18) Line 213, “the Arabidopsis SMO1 and SMO2 proteins” should be “two Arabidopsis SMO1 isoforms”.
19) Line 240, “ABP” should be “APB”.
20) Line 249, “genes of interested” should be “genes of interest”.
21) Line 264, please add reference after “response to auxin”.
22) Line 267, please add reference after “loss-of-function mutants”.
23) Line 278, “Caenorhabditis elegans” should be italicized.
24) Line 294, “regulate” should be “regulates”.
25) Line 304, please add “group” after “one single methyl”.
26) Line 310, “perform” should be “performs”.
27) Line 324, “APB” should be defined in full at the first appearance in Line 239.
28) Line 325 and 326, “ABP” should be “APB”.
29) Line 328, please add “were” after “isolated”.
30) Line 330, please change comma to semi-colon.
31) Line 331, please change comma to “; and”.
32) Line 337, please add period after “fungitoxicity [119]”
33) Line 374, “are low” should be “is low”.
34) Line 395, “were tested” should be “was tested”.
35) Line 396, “in vitro” and “in vivo” should be italicized.
36) Line 452, is “responsible for” more appropriate than “acting on”?
37) Line 452, “in mammal, yeast, plant” should be “in mammals, yeast, plants”.
38) Line 470, is “detectable” more appropriate than “considered”?
Author Response
Reviewer 2
Darnet and Schaller review the current knowledge on the functional roles of C4-methylsterols and properties of the pertinent biosynthetic enzymes. In general, this is a very comprehensive overview which provides readers a clear comparison and contrast between C4-methylsterol functions in mammals, fungi, plants and bacteria. Given the lack of recent reviews on the similar topic, this work is considered a significant contribution to the field.
We would like to thank Reviewer#2 for her/his positive comments on our work.
Some minor comments are listed as follows:
1) Line 11, the first letter should be capitalized (i.e. 4,4-Dimethylsterols). There are many other instances throughout the text and Table 1.
We have checked the capitalization of the first letter of a molecule name, depending on the position (first or not) in the sentence.
2) Line 13, “biochemical inhibitions” should be “biochemical inhibition”.
We have corrected this in the text.
3) Line 33, please add “in plants” after “phytosterols”.
“in plants” was added in the text, line 33.
4) Line 46, “organism” should be “organisms”.
We have corrected this in the text.
5) In Figures 1 and 4, please provide the full names besides abbreviations wherever possible.
We have added the full names wherever possible.
6) Line 53, “Sterol and 4-Methylsterol” should be “Sterol and 4-methylsterol”.
We have corrected this in the text.
7) Line 68, “Methylococcus capsulatus” should be italicized.
We have corrected this in the text.
8) Line 92, “accumulations” should be “accumulation”.
We have corrected this in the text.
9) In Figure 3, it is not clear how the plant photo in the graphical inset is related to auxin signaling without labeling.
We have modified the graphical insets and the graphical.
The graphical insets of Figure 2 have been simplified: the arrows are deleted in order for the reader to visualize just a molecule and a target (protein) or a process (plant growth and height).
The graphical abstract has been extensively modified to answer to this point of criticism. Its title is “Bioactive 4-methylsterols”. Very short sentences have been added like “4-carboxyzymosterol binds the receptor gamma of lymphoid cells” and “Oxojessic acid disrupts auxin-promoted growth". The plant photo has now labels.
10) Line 107, “… from references [91; top panel] and [37; bottom panel]” should be “… from references [91 in top panel and 37 in bottom panel]”.
This is corrected in the figure legend.
11) Line 113-114, “A difference between sterol pathways of mammals and fungi, and plants is …” should be “A difference between sterol pathways of mammals and fungi versus plants is …”.
We have corrected this in the text.
12) Line 141, “C4-carboxy group” should be “C4-carboxyl group”.
We have corrected in the text.
13) In Table 1, the numerous molecules are categorized into M1 to M9. The IDs should be explained.
We have amended Table 1 and molecule numbering.
14) Line 159, “uptake” should not be used as a verb.
We have corrected in the text.
15) Line 164, “a NAD(P)-dependent” should be “an NAD(P)-dependent”.
We have corrected in the text.
16) Line 175, please delete “this”.
We have deleted the word in the text.
17) Line 185, please add a reference after "cell viability".
We have added the corresponding reference.
18) Line 213, “the Arabidopsis SMO1 and SMO2 proteins” should be “two Arabidopsis SMO1 isoforms”.
We have made the substitution.
19) Line 240, “ABP” should be “APB”.
We have updated the text.
20) Line 249, “genes of interested” should be “genes of interest”.
We have corrected in the text.
21) Line 264, please add reference after “response to auxin”.
We have added the reference.
22) Line 267, please add reference after “loss-of-function mutants”.
We have added the reference.
23) Line 278, “Caenorhabditis elegans” should be italicized.
We have corrected this issue in the text.
24) Line 294, “regulate” should be “regulates”.
We have corrected the text.
25) Line 304, please add “group” after “one single methyl”.
We have corrected the text.
26) Line 310, “perform” should be “performs”.
We have corrected in the text.
27) Line 324, “APB” should be defined in full at the first appearance in Line 239.
We have defined APB in full at the first appearance in the text.
28) Line 325 and 326, “ABP” should be “APB”.
We have corrected in the text.
29) Line 328, please add “were” after “isolated”.
We have corrected in the text.
30) Line 330, please change comma to semi-colon.
We have corrected in the text.
31) Line 331, please change comma to “; and”.
We have corrected in the text.
32) Line 337, please add period after “fungitoxicity [119]”
We have corrected in the text.
33) Line 374, “are low” should be “is low”.
We have corrected in the text.
34) Line 395, “were tested” should be “was tested”.
We have corrected in the text.
35) Line 396, “in vitro” and “in vivo” should be italicized.
We have corrected in the text.
36) Line 452, is “responsible for” more appropriate than “acting on”?
We have corrected in the text.
37) Line 452, “in mammal, yeast, plant” should be “in mammals, yeast, plants”.
We have corrected in the text.
38) Line 470, is “detectable” more appropriate than “considered”?
We have corrected in the text.
Reviewer 3 Report
The authors review the biosynthesis and bioactivities of 4-methyl- and 4,4-dimethyl- sterols. They highlight examples from major cladistic kingdoms, including prokaryotes, fungi, plants, and animals These include major and transient 4-methylsterols, as well as non-canonical 4-methylsterols. Important functions of 4-methylsterols, such as oxojessic acid in auxin signaling and 4ACD8 in lymphoid cell development, are discussed. 4-Methylsterols in mutants lacking sterol biosynthetic enzymes and inhibited organisms are also discussed. Overall, the review manuscript is clear, thorough, and organized well. Minor comments are as follows.
In Figure 1A, it seems as though conventional numbering is used instead of IUPAC. However, if the conventional numbering system is intended, C30 and C31 are transposed. Is this meant to be the conventional system?
In Figure 1C (right) the structure does not match the label. The label says 24-ethyl and no hydroxyl, while the structure has 24-methyl and the hydroxyl group.
In several figures, particularly 2 and 3, several fonts are used, and structures do not use the same font sizes and drawing conventions. The instructions for this journal indicate that structures should match from figure to figure.
Line 131, please change methylene to methyl.
Table 1 is missing several methyl groups. E.g. M26, and M31 say “4α-hydroxy” rather than “4α-hydroxymethyl”; M11 says "4-hydroxy" rather than 30 or 31 hydroxy.
Line 307, “This” should be “These”.
Line 328 mentions PF1163A and PF1163B in Figure 4. Was it your intention to include PF1163B in Figure 4?
Line 4,4-dimethylzymosterol is mentioned, though Table 1, Figure 1, and elsewhere in the manuscript identify this as T-MAS. It may be helpful to include 4,4-dimethylzymosterol as an alternate name on Table 1.
Author Response
Reviewer 3
The authors review the biosynthesis and bioactivities of 4-methyl- and 4,4-dimethyl- sterols. They highlight examples from major cladistic kingdoms, including prokaryotes, fungi, plants, and animals These include major and transient 4-methylsterols, as well as non-canonical 4-methylsterols. Important functions of 4-methylsterols, such as oxojessic acid in auxin signaling and 4ACD8 in lymphoid cell development, are discussed. 4-Methylsterols in mutants lacking sterol biosynthetic enzymes and inhibited organisms are also discussed. Overall, the review manuscript is clear, thorough, and organized well. Minor comments are as follows.
We appreciate the positive comments of Reviewer#3 on our manuscript.
In Figure 1A, it seems as though conventional numbering is used instead of IUPAC. However, if the conventional numbering system is intended, C30 and C31 are transposed. Is this meant to be the conventional system?
We have corrected the carbon numbering, using the IUPAC system for sterol and steroids (Moss, G. P., Nomenclature of steroids (Recommendations 1989). Pure Appl. Chem. 1989, 61, (10), 1783-1822.)
In Figure 1C (right) the structure does not match the label. The label says 24-ethyl and no hydroxyl, while the structure has 24-methyl and the hydroxyl group.
We thank Reviewer#3 for detecting this mistake. We have corrected the side chain structure.
In several figures, particularly 2 and 3, several fonts are used, and structures do not use the same font sizes and drawing conventions. The instructions for this journal indicate that structures should match from figure to figure.
We have redrawn all the final figures with the same conventions in ChemDraw, according to the author instructions of the journal.
Line 131, please change methylene to methyl.
The substitution was done in the text.
Table 1 is missing several methyl groups. E.g. M26, and M31 say “4α-hydroxy” rather than “4α-hydroxymethyl”; M11 says "4-hydroxy" rather than 30 or 31 hydroxy.
Table 1 was updated, and 4α position nomenclature was checked.
Line 307, “This” should be “These”.
The sentence was corrected (line 307).
Line 328 mentions PF1163A and PF1163B in Figure 4. Was it your intention to include PF1163B in Figure 4?
The PF1163B structure was added in Figure 5.
Line 4,4-dimethylzymosterol is mentioned, though Table 1, Figure 1, and elsewhere in the manuscript identify this as T-MAS. It may be helpful to include 4,4-dimethylzymosterol as an alternate name on Table 1.
We have included the 4,4-dimethylzymosterol as an alternate name in Table 1, and thus in yeast, the T-MAS name is not used.